# A Novel Method to Analyze the Relationship between Thermoelectric Coefficient and Energy Disorder of Any Density of States in an Organic Semiconductor

**DOI:** 10.3390/mi14081509

**Published:** 2023-07-27

**Authors:** Dong Qin, Jiezhi Chen, Nianduan Lu

**Affiliations:** 1School of Information Science and Engineering, Shandong University, Qingdao 266237, China; dongqin@mail.sdu.edu.cn (D.Q.); chen.jiezhi@sdu.edu.cn (J.C.); 2The State key Lab of Fabrication Technologies for Integrated Circuits & Laboratory of Microelectronics Devices and Integrated Technology, Institute of Microelectronics, Chinese Academy of Sciences, Beijing 100029, China

**Keywords:** density of states, organic semiconductors, Seebeck coefficient, degree of disorder

## Abstract

In this work, a unified method is proposed for analyzing the relationship between the Seebeck coefficient and the energy disorder of organic semiconductors at any multi-parameter density of states (DOS) to study carrier transport in disordered thermoelectric organic semiconductors and the physical meaning of improved DOS parameters. By introducing the Gibbs entropy, a new multi-parameter DOS and traditional Gaussian DOS are used to verify this method, and the simulated result of this method can well fit the experiment data obtained on three organic devices. In particular, the impact of DOS parameters on the Gibbs entropy can also influence the impact of the energy disorder on the Seebeck coefficient.

## 1. Introduction

Organic semiconductors for thermoelectric applications have been widely studied and applied in recent years due to their low thermal conductivity, molecular diversity, non-toxicity, etc. [1,2,3]. The Seebeck coefficient is an important parameter of the electrical properties of thermoelectric materials used to characterize the size of the Seebeck effect [4]. Moreover, as the size of organic semiconductor devices becomes smaller and smaller, the interface contact effect plays an increasingly important role in charge transport. Seebeck voltage is not affected by ithe nterface contact, so the Seebeck effect is a new way to reveal carrier transport [5]. Therefore, models for calculating the Seebeck coefficient of organic semiconductors have been extensively studied.

As it is well known, the Gaussian disorder model (GDM) is the most commonly used model to describe carrier transport in disordered organic semiconductors, and its expression for DOS is $g\left(E\right)=\frac{N_{t}}{\sqrt{2\pi }\sigma_{DOS}}\exp\left(-\frac{E^{2}}{2\sigma_{DOS}^{2}}\right)$ [6,7,8,9,10,11]. Considering the diversity of organic semiconductor structures and the limitations of GDM at high concentration, various improved DOS models have been proposed and used to make up for the disadvantages of GDM, such as exponential DOS, DOS based on free probability approximation, etc. [12,13,14,15,16]. In addition, doping, as a common means to improve the performance of organic thermoelectric materials, will also change the DOS and the band gap of organic semiconductors [17,18,19]. Hence, several DOS superimposed models or completely new models have been used to study the doping process of organic semiconductors [8,20,21,22,23]. It is well known that the degree of disorder is an important parameter of disordered organic semiconductors, reflecting the structural characteristics of organic semiconductors to some extent [24,25]. Doping can affect the degree of disorder, which is closely related to the characteristics of DOS and the Seebeck coefficient [26,27]. However, since these newly proposed DOS models are superimposed by multiple traditional densities of states or have multiple parameters, their parameters cannot directly reflect the disorder as the $\sigma_{DOS}$ of GDM. Thus, in order to study the effect of new model disorder on the Seebeck coefficient and the physical meaning of new DOS parameters, a general method is needed to analyze the relationship between the disorder of various DOS models and the Seebeck coefficient.

In this work, by introducing the Gibbs entropy, a general method is presented to calculate the disorder-dependent Seebeck coefficient of the traditional and improved DOS models in an organic semiconductor, which can be applied to any suitable DOS. This method can be verified by a traditional Gaussian DOS model through the comparison of simulation data and experimental data. In addition, a new multi-parameter improved DOS model is also proposed to calculate the Seebeck coefficient of the organic semiconductor. By calculation analysis, it can be used to validate this method as well.

## 2. Model Theory

In general, taking the p-type as an example, one can calculate the Seebeck coefficient (*S*) of disordered organic semiconductors via the following equation [28]:(1)$$S=-\frac{k_{B}}{q}\int_{-\infty }^{\infty }\frac{E-E_{F}}{k_{B}T}\frac{\sigma (E)}{\sigma }dE=-\frac{1}{qT\sigma }\int_{-\infty }^{\infty }E\sigma \left(E\right)dE+\frac{E_{F}}{qT},$$
where $k_{B}$ is the Boltzmann constant, *q* is the unit charge, *T* is the temperature, *E* is the energy, $E_{F}$ is the Fermi energy and σ is the electrical conductivity. Obviously, $\sigma =\int_{-\infty }^{\infty }\sigma \left(E\right)dE$.

Then, according to the carrier concentration equation, the relationship between the Fermi energy ($E_{F}$) and the carrier concentration (*n*) can be expressed as [15]
(2)$$n=\int_{-\infty }^{\infty }g\left(E\right)f\left(E\right)dE,$$
where f(E) is the Fermi–Dirac distribution that is represented by equation $f\left(E\right)={\left\{1+\exp\left[\left(E_{F}-E\right)/k_{B}T\right]\right\}}^{-1}$ [16]. g(E) represents the density of states (DOS) of disordered organic semiconductors. To reflect the universality of this method, any suitable g(E) can be used in the DOS function.

As we mentioned, exponential DOS and Gaussian DOS are the two most commonly used densities of states in organic semiconductors [29,30]. The DOS of organic semiconductors is still unclear even though the exponential and the Gaussian models have been proposed to simulate the DOS tails going into the energy gap, which is also called the “Urbach tail” [31,32]. Up to now, there was no direct experimental evidence verifying specific DOS models of organic semiconductors since the variations between various experiments were higher than the difference between the theoretical approaches. Although the two DOS work well under different situations, their limitations are also evident. For example, when calculating mobility, the exponential DOS does not work at low concentrations, and the Gaussian DOS does not work at high concentrations [33]. More importantly, as stated by some researchers, the Gaussian DOS is only an assumption and may not strictly be true, and the DOS of disordered organic semiconductors cannot be purely exponential at a low concentration [16,29]. Therefore, modification on Gaussian DOS and exponential DOS is a common method to deal with the DOS of disordered organic semiconductors.

In this work, based on Gaussian DOS and exponential DOS, we proposed a new multi-parameters function used to better demonstrate this method (see Appendix A for the source of this new DOS):(3)$$g\left(E\right)=\left\{\begin{array}{cc} \frac{N_{t}}{\alpha^{-1}\beta }{\left(\frac{E}{\beta }\right)}^{\alpha -1}\exp\left[-{\left(\frac{E}{\beta }\right)}^{\alpha }\right] & E\geq 0\\ 0 & E<0 \end{array}\right.,$$
where $N_{t}$ is the number of states per unit volume. α and β are the proposed DOS parameters, and their effect on the proposed DOS is shown in Figure 1. α is the shape parameter of DOS and it controls the proportional distribution of the extended and localized states. β is the width parameter of DOS and it has a larger effect on the width of DOS, just like $\sigma_{DOS}$ of Gaussian DOS.

Firstly, this new multi-parameter DOS is a modification of purely exponential DOS. As shown in Figure 1a and Equation (Equation 3), it is not difficult to see that this proposed DOS degenerates into exponential DOS when α=1. In addition, the value of α in the interval of 1 to 2 can make the tail convergence degree of DOS between exponential DOS and Gaussian DOS, which is similar to the result obtained by Oelerich et al. [16]. Finally, α can also represent the degree of doping of organic semiconductors to a certain extent. Nowadays, organic semiconductors are more or less doped for higher mobility. As stated by some researchers, with the increase in the doping degree of organic semiconductors, the DOS tail of an organic semiconductor becomes heavier and the Gaussian peak value shifts. Obviously, as shown in Figure 1a, the smaller α, the heavier the tail of the DOS and the more severe the Gaussian peak shift, that is, the heavier the doping of organic semiconductors, which is consistent with the effect of DOS in the previous articles [22,23]. In addition, according to our previous work, one could obtain the relationship between the parameters (α and β) and the intrinsic properties of these organic materials by equilibrium energy theory and transport energy theory [34].

To calculate the conductivity, based on the common Miller–brahams hopping model [21,35] one can obtain the average hopping rate v(E) at which carriers transition from energy *E* to a different energy. It is given by the sum of the average downward ($v_{↓}\left(E\right)$) and upward ($v_{↑}\left(E\right)$) hopping rates [25]:(4)$$v_{↓}\left(E\right)=v_{0}\exp\left[-2\alpha_{0}R\left(E\right)\right],$$
(5)$$v_{↑}\left(E\right)=\int_{-\infty }^{E}\exp\left(-\frac{E-\varepsilon }{k_{B}T}\right)v_{↓}\left(\varepsilon \right)\left(\frac{g(\varepsilon )[1-f(\varepsilon )]}{\int_{-\infty }^{E}g\left(\varepsilon \right)\left[1-f\left(\varepsilon \right)\right]d\varepsilon }\right)d\varepsilon ,$$
where $v_{0}$ is the attempt-to-jump frequency ($v_{0}≃1.0\times 10^{12}\;s^{-1}$), $\alpha_{0}$ is the inverse localization radius, R(E) is the average distance carriers at energy *E* and it is expressed as $R\left(E\right)={\left\{\frac{4\pi }{3B}\int_{E}^{\infty }g\left(\varepsilon \right)\left[1-f\left(\varepsilon \right)\right]d\varepsilon \right\}}^{-\frac{1}{3}}$ [13]. Here, the percolation threshold parameter B≃2.7. Then, by substituting Equations (4) and (5) into the generalized Einstein relation [36], the carrier mobility (μ) at energy *E* can be obtained: (6)$$\begin{array}{cc} \; & \mu \left(E\right)=\frac{q[1-f(E)]}{k_{B}T}\left[R{\left(E\right)}^{2}v_{↓}\left(E\right)+\right.\\ \; & \left.\int_{-\infty }^{E}R{\left(\varepsilon \right)}^{2}\exp\left(-\frac{E-\varepsilon }{k_{B}T}\right)v_{↓}\left(\varepsilon \right)\left(\frac{g(\varepsilon )[1-f(\varepsilon )]}{\int_{-\infty }^{E}g\left(\varepsilon \right)\left[1-f\left(\varepsilon \right)\right]d\varepsilon }\right)d\varepsilon \right]. \end{array}$$

As is well known, the relationship between mobility μ(E) and conductivity σ(E) can be expressed as follows [3]:(7)σ(E)=qg(E)f(E)μ(E).

Obviously, by substituting Equation (Equation 6) into Equation (Equation 7), one can obtain σ(E). And then, by substituting σ(E) and $E_{F}$ into Equation (Equation 1), the Seebeck coefficient (*S*) can be obtained.

Next, we solve the problem of how to express the disorder of different DOS, and additionally solve the problem of the physical meaning of the DOS parameters. Generally speaking, the disorder of a system is closely related to the entropy of this system. Physically, the entropy means the disorder or the degree of chaos in a system. That means the more disordered the system is, the higher the entropy value is. As the variation of Boltzmann’s entropy, the Gibbs entropy ($S_{GE}$) has been effectively used in some areas of statistical physics and statistics. It can be given by the following equation [37]:(8)$$S_{GE}=k_{B}\ln\Omega =-k_{B}\sum_{i}p_{i}\ln p_{i},$$
where Ω is the number of microscopic states, and $p_{i}$ is the probability of microscopic states.

As is well known, microscopic particles have wave–particle duality. In addition, the energy levels of disordered organic semiconductors are discrete rather than continuous, which is consistent with the precondition for using the Gibbs entropy. To ensure $S_{GE}$ and DOS to be better described quantitatively rather than qualitatively, one can assume that the energy levels of disordered organic semiconductors are $\left\{E_{1},E_{2},\ldots ,E_{i},\ldots \right\}$ with the energy interval ΔE. Based on the definition of DOS that it is a statistical description of the number of quantum states at energy *E* and the basic concept of probability density in statistics, one can reasonably assume that the probability of occupation at each state is equal. Then, one can obtain the following equation:(9)$$p_{i}=\frac{N_{i}}{\sum_{i}N_{i}}=\frac{N_{i}}{N}=\frac{g\left(E_{i}\right)\Delta E}{N},$$
where $N_{i}$ means the number of microscopic states between energy level $E_{i}$ and $E_{i+1}$, and *N* means the total number of microscopic states.

Obviously, by substituting g(E) and Equation (Equation 9) into Equation (Equation 8), one can obtain the Gibbs entropy of DOS. In other words, this establishes the relationship between the DOS and the degree of disorder, regardless of the type of DOS. In this work, we take the DOS of Equation (Equation 3) as an example to illustrate it better. By this means, one can obtain
(10)$$S_{GE}=k_{B}\left[\ln\frac{\beta }{\Delta E}-\ln\alpha +\frac{(\alpha -1)\gamma }{\alpha }+1\right],$$
where γ is Euler’s constant and it is equal to 0.5772…

Then, one can use its partial derivative to further illustrate the influence of each parameter on the Gibbs entropy, which are $\frac{\partial S_{GE}}{\partial \alpha }=\frac{k_{B}\left(\gamma -\alpha \right)}{\alpha^{2}}$ and $\frac{\partial S_{GE}}{\partial \beta }=\frac{k_{B}}{\beta }$. Obviously, the Gibbs entropy increases as β increases, which means that the increase in β reflects the increase in the disorder of an organic semiconductor. On the contrary, the degree of disorder of an organic semiconductor decreases as α increases (α>γ).

## 3. Results and Discussion

First, we verified the rationality and feasibility of this proposed model by comparing the simulation result with the experimental data [38]. Figure 2a shows that the simulation results can well fit the concentration-dependent experimental results of the Seebeck effect. (The other fitting parameters are IDTBT: $N_{t}=7.4\times 10^{20}\;{\mathrm{cm}}^{-3}$, $\alpha_{0}^{-1}=0.6\;\mathrm{nm}$; PBTTT: $N_{t}=2.0\times 10^{20}\;{\mathrm{cm}}^{-3}$, $\alpha_{0}^{-1}=0.35\;\mathrm{nm}$; PSeDPPBT: $N_{t}=3.2\times 10^{22}\;{\mathrm{cm}}^{-3}$, $\alpha_{0}^{-1}=0.6\;\mathrm{nm}$.) With the increase in carrier concentration, the transport energy ($E_{t}$) and the Fermi energy ($E_{F}$) become closer, which leads to the decrease in the Seebeck coefficient. Obviously, the slopes of the Seebeck coefficient of the three devices are different. By calculating the Gibbs entropy ($S_{GE}$) of Equation (Equation 10) corresponding to the DOS of the three devices, it can be found that the Gibbs entropy of IDTBT is the lowest and the Gibbs entropy of PSeDPPBT is the highest. Therefore, PSeDPPBT has the highest degree of disorder and the widest DOS. Consequently, when the concentration of PSeDPPBT changes, the difference between $E_{t}$ and $E_{F}$ changes the most, and then the Seebeck coefficient changes the most, that is, the curve tilts the steepest. In addition, as stated by some researchers, the Seebeck effect is essentially a thermal excitation process determined by entropy transport. In other words, when the width of DOS is relatively small (the smaller width parameter β, the smaller the Gibbs entropy $S_{GE}$), the transport state of the thermal excitation also becomes small so that the slope of the Seebeck coefficient with the concentration is less affected by temperature. It can be seen that our model theory can well explain the experimental phenomenon.

At the same time, we also compared the temperature-dependent simulation results of the Seebeck effect with the experimental results of pentacene films on two different substrates [39], as shown in Figure 2b. (The other fitting parameters are pentacene films on Si/SiO${}_{2}$ substrate: $N_{t}=3.0\times 10^{20}\;{\mathrm{cm}}^{-3}$, $\alpha_{0}^{-1}=1\;\mathrm{nm}$; pentacene films on Si/SiO${}_{2}$/Al${}_{2}$O${}_{3}$ substrate: $N_{t}=3.0\times 10^{20}\;{\mathrm{cm}}^{-3}$, $\alpha_{0}^{-1}=0.24\;\mathrm{nm}$.) It can be seen that the Seebeck coefficient decreases slightly with the increase in temperature. The reason is that when the temperature of the pentacene films is higher, the energy of the carrier is higher, and the carrier more easily participates in the transmission through thermal excitation, which leads to an increase in conductivity of the pentacene films, so the Seebeck coefficient is reduced. Similarly, the Gibbs entropy calculation shows that the Gibbs entropy of the pentacene films on the Si/SiO${}_{2}$/Al${}_{2}$O${}_{3}$ substrate is higher than that of the pentacene films on the Si/SiO${}_{2}$ substrate. Therefore, the energy disorder of the pentacene films on the Si/SiO${}_{2}$/Al${}_{2}$O${}_{3}$ substrate is larger, which causes their Seebeck coefficients to also become more affected by temperature.

Next, in order to further analyze the relationship between the degree of disorder and the Seebeck coefficient, one can plot the influence of the proposed DOS parameters on the Seebeck coefficient at different temperatures and concentrations. As shown in Figure 3e,f, when α decreases or β increases, the Gibbs entropy ($S_{GE}$) increases, that is, the degree of disorder increases. Then, from Figure 3a–d, at the same temperature and concentration, the larger the degree of disorder, the larger the Seebeck coefficient. This is because the greater the disorder, the larger the difference between $E_{t}$ and $E_{F}$ (the greater the activation energy), so the larger the Seebeck coefficient. What is more, the influence of concentration and temperature on teh Seebeck coefficient also becomes larger (the slope of the curve increases) with the disorder increasing. This is consistent with our previous analysis of the experimental results in Figure 2a. In addition, by comparing Figure 3e,f, it can be found that the change in β obviously causes a larger change in the Gibbs entropy compared with α. That is why the curves in Figure 3b,d change more than the curves in Figure 3a,c.

Then, in order to express the effect of disorder on the Seebeck coefficient more directly, one can also plot the Seebeck coefficient (*S*) as a function of the Gibbs entropy ($S_{GE}$). As shown in Figure 4, converted from the data in Figure 3c, the Seebeck coefficient increases with the degree of disorder increasing (Gibbs entropy increasing). Moreover, the lower the temperature, the greater the Seebeck coefficient increase with the degree of disorder. A similar result was also obtained by Lu et al. based on the Gaussian DOS and the percolation theory [40], which shows that the method in this work is universal tfor any density of states. Similarly, the relationship between the Seebeck coefficient and the Gibbs entropy at different concentrations can also be obtained through converting the data in Figure 3a,b.

Finally, in order to better explain that our proposed model could also be applied to other reasonable DOS, we simulated the Seebeck effect based on Gaussian DOS, which is the most commonly used DOS model. By using the same simulation method, the comparison between the Seebeck coefficient calculation results based on Gaussian DOS and the experimental data is shown in Figure 5. (The other fitting parameters are IDTBT: $N_{t}=7.4\times 10^{20}\;{\mathrm{cm}}^{-3}$, $\alpha_{0}^{-1}=0.6\;\mathrm{nm}$; PBTTT: $N_{t}=1.5\times 10^{20}\;{\mathrm{cm}}^{-3}$, $\alpha_{0}^{-1}=0.37\;\mathrm{nm}$; PSeDPPBT: $N_{t}=7.0\times 10^{22}\;{\mathrm{cm}}^{-3}$, $\alpha_{0}^{-1}=0.75\;\mathrm{nm}$.) Obviously, the larger the width parameter $\sigma_{DOS}$ of Gaussian DOS, the more Seebeck coefficient is affected by the change in concentration, and the steeper the slope of the Seebeck coefficient curve.

Similarly, by substituting Gaussian DOS $g\left(E\right)=\frac{N_{t}}{\sqrt{2\pi }\sigma_{DOS}}\exp\left(-\frac{E^{2}}{2\sigma_{DOS}^{2}}\right)$ and Equation (Equation 9) into Equation (Equation 8), the Gibbs entropy expression of Gaussian DOS can be obtained as follows:(11)$$S_{GE}=k_{B}\ln\frac{\sqrt{2\pi e}\sigma_{DOS}}{\Delta E},$$
where *e* is the Euler number and it is equal to 2.718…

According to Equation (Equation 11), the larger the width parameter $\sigma_{DOS}$, the larger the Gibbs entropy, which is consistent with our previous result (the Gibbs entropy of IDTBT is the lowest and the Gibbs entropy of PSeDPPBT is the highest). Hence, the larger the width parameter $\sigma_{DOS}$ of DOS, the greater the energy disorder, which leads to the Seebeck coefficient being more affected by the concentration. It should be emphasized that some previous articles default $\sigma_{DOS}$ to represent the energy disorder when using the Gaussian DOS model [7,8,9,29]. This is only a qualitative description based on experience, while our proposed method can offer a quantitative description to the energy disorder of Gaussian DOS.

In addition, one can also plot the influence of the Gaussian DOS width parameter $\sigma_{DOS}$ on the Seebeck coefficient at different temperatures and concentrations. Figure 6 shows that the Seebeck coefficient changes with temperature and concentration for different width parameter $\sigma_{DOS}$ of Gaussian DOS. It can be seen that under the same conditions, the larger $\sigma_{DOS}$, the larger the energy disorder, and then the larger the Seebeck coefficient. What is more, whether the Seebeck coefficient changes with concentration in Figure 4a or it changes with temperature in Figure 4b, the larger $\sigma_{DOS}$, the more they change (the steeper the slope). The phenomenon in Figure 6 is consistent with our previous analysis in Figure 3 and Figure 4, which further proves the usability of our proposed Seebeck model and the validity of the analysis of the relationship between the Gibbs entropy and the energy disorder of an organic semiconductor.

Of course, one can also directly plot the relationship between the Seebeck coefficient and the Gibbs entropy of Gaussian DOS based on simulation data in Figure 6 and Equation (Equation 11). The relationship between the Gibbs entropy and width parameter $\sigma_{DOS}$ of Gaussian DOS is shown in Figure 7a, and the relationship between the Gibbs entropy and the Seebeck coefficient is shown in Figure 7b. As it can be seen from Figure 7b, the Seebeck coefficient increases with the increase in the Gibbs entropy (energy disorder). Moreover, the lower the temperature, the greater the slope of the Seebeck coefficient growth, indicating that the lower the temperature, the greater the dependence of the Seebeck coefficient on the disorder. This is also consistent with our previous analysis in Figure 4.

## 4. Conclusions

We proposed a general calculation method to express the relationship between the Seebeck coefficient and the energy disorder at any different multi-parameter organic semiconductor densities of states by introducing the Gibbs entropy. Through the comparison of simulation data and experimental data with the Seebeck coefficient varying in temperature and concentration, this new proposed method is applicable since the simulated results agree well with the measured experimental data. In addition, by analyzing the calculation result, this proposed method is suitable for not only Gaussian DOS, but also for the new multi-parameter DOS proposed in this work. Therefore, the universality of the proposed method is also verified. This method provides impressive potential for future investigation on doping or improved DOS model.

## Figures and Tables

**Figure 1 micromachines-14-01509-f001:**
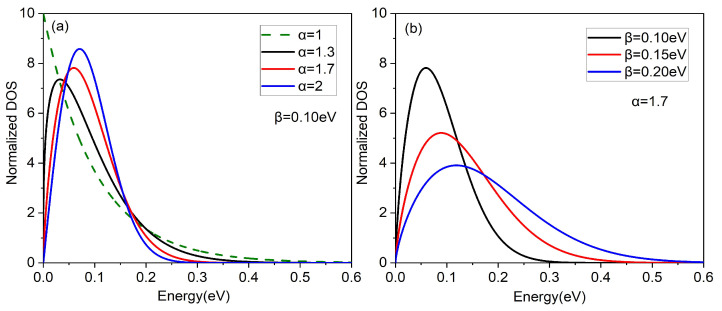
(**a**) The proposed DOS for different parameters α and β=0.1eV. (**b**) The proposed DOS for different parameters β and α=1.7.

**Figure 2 micromachines-14-01509-f002:**
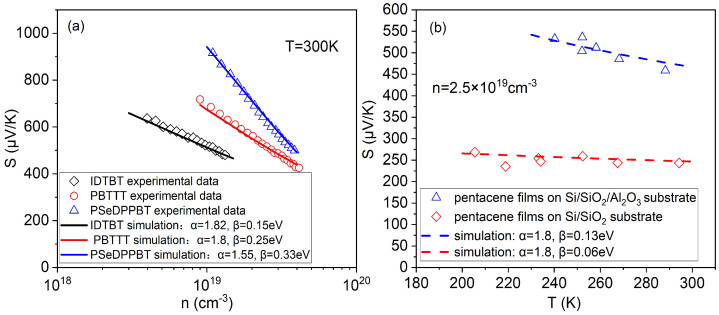
(**a**) Seebeck coefficient simulated with the change in the charge carrier concentration at T=300K for three different devices compared with experimental data [38]; (**b**) Seebeck coefficient simulated with the change in the temperature at charge carrier concentration $n=2.5\times 10^{19}\;{\mathrm{cm}}^{-3}$ for pentacene films on two different substrates compared with experimental data [39].

**Figure 3 micromachines-14-01509-f003:**
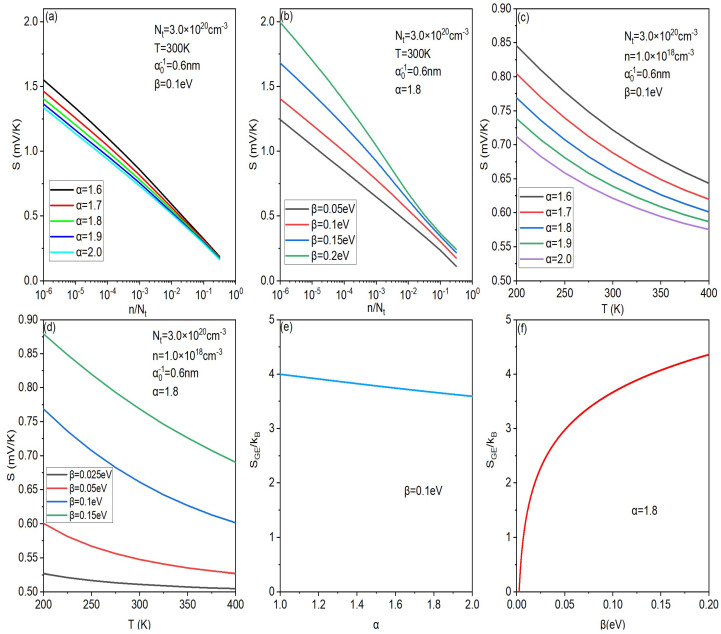
Carrier concentration contribution of the Seebeck effect at different α (**a**) and different β (**b**). Temperature contribution of the Seebeck effect at different α (**c**) and different β (**d**). Gibbs entropy as a function of parameters α (**e**) and β (**f**).

**Figure 4 micromachines-14-01509-f004:**
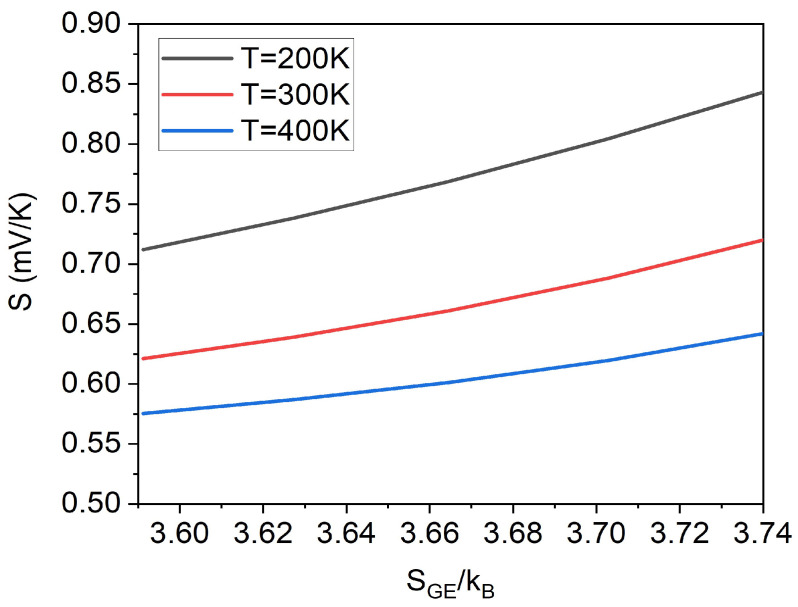
The Seebeck effect (*S*) as a function of Gibbs entropy ($S_{GE}$) at different temperatures, converted from the data in Figure 3c.

**Figure 5 micromachines-14-01509-f005:**
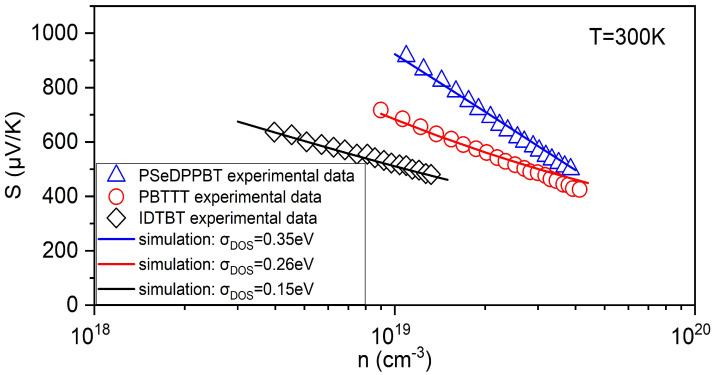
Seebeck coefficient simulated based on Gaussian DOS with the change in the charge carrier concentration at T=300K for three different devices, comparing with experimental data.

**Figure 6 micromachines-14-01509-f006:**
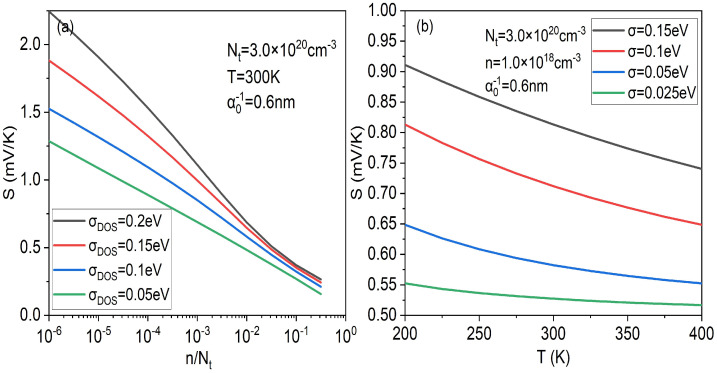
(**a**) Carrier concentration contribution of the Seebeck effect at different width parameter $\sigma_{DOS}$ of Gaussian DOS. (**b**) Temperature contribution of the Seebeck effect at different width parameter $\sigma_{DOS}$ of Gaussian DOS.

**Figure 7 micromachines-14-01509-f007:**
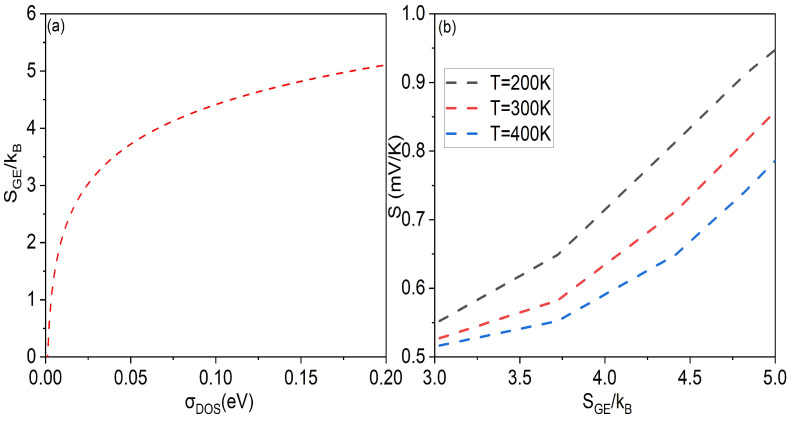
(**a**) Gibbs entropy ($S_{GE}$) of Gaussian DOS as a function of width parameter $\sigma_{DOS}$ of Gaussian DOS; (**b**) the Seebeck effect (*S*) as a function of Gibbs entropy ($S_{GE}$) of Gaussian DOS at different temperatures, converted from the data in Figure 6b.

## Data Availability

Not applicable.

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
