# Peer review of "A Novel Method to Analyze the Relationship between Thermoelectric Coefficient and Energy Disorder of Any Density of States in an Organic Semiconductor"

_micromachines, 2023, doi:10.3390/mi14081509_

Round 1

Reviewer 1 Report

This is a confusing manuscript with claims that are not fully supported by the data. First of all, what is the rationale of equation 3? What is the underlying physical meaning? Are their data supporting such form?

Does equation 9 assume an equal probability of occupation at each state? This is confusing.

A good fit alone does not necessarily validate the theory, especially when considering that the data points are few. I would not be surprised if using another arbitrary form of DOS function can also lead to good agreements. To show the model is correct, results from another method are needed to corroborate the claim: “the Gibbs entropy of IDTBT is the lowest and the Gibbs entropy of PSeDPPBT is the highest”.

Many statements need clarification. The authors should clarify the meaning of “but also can be used itself” on page 7.

needs to be improved

Reviewer 2 Report

This study provides a new method capable of explaining and understanding the carrier transport properties in organic semiconductors, and is expected to be used to design new materials with special properties. The new method proposed in this paper can be used to calculate the chaos-dependent Seebeck coefficients of any modified DOS model and can be used with any appropriate DOS. The model demonstrates its accuracy and robustness in terms of computation and validation. Although the formula reasoning seems to be correct, it lacks experimental support. Can you provide experimental data support? 

good 

Reviewer 3 Report

The manuscript «A novel method to analyze the relationship between thermoelectric coefficient and disorder of any density of states in organic semiconductor» by Dong Qin, Jiezhi Chen and Nianduan Lu is devoted to clarifying the fundamental question of the physics of disordered organic materials - the relationship between the Seebeck coefficient and disorder. The measure of disorder is the Gibbs entropy. The relationship between the entropy and the DOS function is found, and it is shown that the Seebeck coefficient increases with increasing disorder. The DOS functions corresponding to the experimental dependences of the Seebeck coefficient on the concentration of charge carriers were used. The results of the work open up new possibilities for experimental determination of the DOS function and modeling of thermoelectric phenomena in various organic materials. I recommend publishing this work in a journal “Micromachines”.

Author Response

Thank you for your kindly review.

Reviewer 4 Report

In this work, the authors proposed a phenomenological, multi-parameter model for density of states (DOS) in disordered organic semiconductors. The two key parameters, α and β, were related to the Urbach tail decay rate and electronic bandwidth, thus holding physical meanings instead of simple fitting parameters. The authors then applied the proposed model in fitting experimental Seebeck coefficients and carrier mobilities of organic semiconductor devices using different polymers. The Gibbs entropy were also computed using the model DOS, and its relation with disorder was discussed.

Overall, the proposed DOS is an improved model over simple exponential or Gaussian DOS tail, in the sense that it has more flexibility while retaining physically meaningful parameters. It is therefore useful in analyzing and understanding experimental charge transport data in organic materials and devices.

I would recommend the article for publication in Micromachines, provided that the following issues are addressed in a revised version.

1.     The distribution in Eqn (3) actually has a name (Weibull distribution) in statistics.

2.     Since the parameters were extracted from fitting, the authors should present the fitting performance, as compared to exponential and Gaussian DOS. Key statistical quantities such as R2 and RMSE should be provided.

3.     The disorders in organic semiconductors (IDTBT, PBTTT, PSeDPPBT, pentacene) have been discussed. The authors should also mention how these parameters (α and β) are related to the intrinsic properties of these materials.

 The language should be improved. For example, I suppose the authors means to say “impact of …” instead of “influence degree of …” in line 6-7; “most commonly used” instead of “most common used”; some sentenses should shortened and rephrased for clarification; etc.

Round 2

Reviewer 1 Report

My comments were mainly about the scientific soundness of the manuscript. Based on the visible portion in the revised manuscript and the authors' reply, these issues remain unsolved. The added Gaussian DOS fits well, which raises more concerns about the necessity of using such an arbitrary form in Eq. 3.

can be improved
